# Cooperative Cu/azodiformate system-catalyzed allylic C–H amination of unactivated internal alkenes directed by aminoquinoline

Le Wang[1], Cheng-Long Wang[1], Zi-Hao Li[1], Peng-Fei Lian[1], Jun-Chen Kang[1], Jia Zhou[1], Yu Hao[1], Ru-Xin Liu[1], He-Yuan Bai [1] & Shu-Yu Zhang [1] ✉

Aliphatic allylic amines are common in natural products and pharmaceuticals. The oxidative intermolecular amination of $C(sp^3)$-H bonds represents one of the most straightforward strategies to construct these motifs. However, the utilization of widely internal alkenes with amines in this transformation remains a synthetic challenge due to the inefficient coordination of metals to internal alkenes and excessive coordination with aliphatic and aromatic amines, resulting in decreasing the reactivity of the catalyst. Here, we present a regioselective Cu-catalyzed oxidative allylic $C(sp^3)$-H amination of internal olefins with azodiformates to these problems. A removable bidentate directing group is used to control the regiochemistry and stabilize the π-allyl·metal intermediate. Noteworthy is the dual role of azodiformates as both a nitrogen source and an electrophilic oxidant for the allylic C–H activation. This protocol features simple conditions, remarkable scope and functional group tolerance as evidenced by >40 examples and exhibits high regioselectivity and excellent *E/Z* selectivity.

Aliphatic amines serve as key structural motifs in natural products, drugs, and pharmaceuticals[1]. Allylic amines are especially attractive synthetic targets as they possess diverse bioactivities and also represent an outstanding regimen of prescription medicines[2], including omacetaxine, pentazocine and afatinib (Fig. 1). Therefore, the development of effective allylamine synthesis methods has been a long-standing goal[3]. One traditional approach toward allylic amines relies on nucleophilic displacement using prefunctionalized alkenes[4], while another strategy utilizes C–H insertion by a metal nitrenoid[5]. More recently, oxidative allylic amination has been recognized as an attractive alternative to assemble allylic amines[6–8].

Selective C–H oxidation reactions have emerged as powerful methods for rapidly introducing functional groups into preformed carbon skeletons[9–13]. In particular, palladium-catalyzed allylic C–H

bond oxidation reactions have been extensively investigated (Fig. 2a)[14–21]. In a seminal report, White and colleagues reported the catalytic allylic C–H cleavage of terminal olefins catalyzed by Pd(II)/bis-sulfoxide[22]. Recently, Trost and Gong's group discovered that phosphorus-based ligands were able to promote palladium mediated allylic C–H activation via a concerted proton and two-electron transfer process[23–26]. However, transition-metal catalyzed allylic C–H oxidation was mostly reported to occur on simple alkenes with cyclic or terminal double bonds. A general and selective allylic oxidation using the simple internal alkenes remains elusive (Fig. 2b)[27–29], presumably because the substrates of terminal alkenes contain only one set of allylic protons. In contrast, internal alkenes possess two sets of protons on either side of the alkene functional group, making the regioselectivity more intricate. In addition, when the resulting product is an internal alkene,

[1]Shanghai Key Laboratory for Molecular Engineering of Chiral Drugs, School of Chemistry and Chemical Engineering, & Key Laboratory of Green and High-End Utilization of Salt Lake Resources, Shanghai Jiao Tong University, Shanghai 200240, PR China. ✉e-mail: zhangsy16@sjtu.edu.cn

**Fig. 1 | Selected bioactive compounds containing allylamine.**

the issue of *E/Z* selectivity remains. On the other hand, inefficient coordination of the metal to internal alkenes and excessive coordination with aliphatic and aromatic amines can decrease the reactivity of the catalyst, which is another challenge of allylic C−H amination. Here, we describe a catalytic intermolecular allylic amination strategy for unactivated internal alkene substrates, wherein a removable bidentate directing group enables regioselectively allylic C−H cleavage and stabilizes the π-allyl-metal intermediate (Fig. 2c).

Over the past few years, various directing groups (DG) have been employed to facilitate difunctionalization of unactivated alkenes, which have made considerable progress[30–35]. To the best of our knowledge, intermolecular metal-catalyzed remote allylic C−H oxidation of internal alkenes by DG has not been previously reported. At the outset, we recognized the necessity to prevent the emergence of free radicals (R·) or nucleophiles (Nu⁻) in the reaction system, otherwise it is likely to be employed to facilitate the addition of double bonds via radical addition, nucleometalation, or migratory insertion to form a kinetically favored metallacycle intermediate by DG, which is further protonated or functionalized to give a monofunctionalized product or 1,2-difunctionalized product. We hypothesized the ideal catalytic platform would employ an electrophilic oxidant (E⁺) proficient in promoting allylic C−H cleavage, thus circumventing the challenges of alkene difunctionalization. Given our continuous interest in the development of C−N bond formation using azodicarboxylates[36–39], these versatile electrophilic reagents have played crucial roles in various organic synthesis applications, including the Mitsunobu reaction, electrophilic amination, radical coupling. Besides these, azodicarboxylates can be used as oxidants in transition metal-catalyzed reactions[40]. We envisioned the use of azodicarboxylates as suitable oxidants to facilitate allylic C−H cleavage and a nitrogen source for constructing C−N bonds.

## Results

### Reaction condition optimization

To test the above-mentioned hypothesis, 3-butenoic acid derivative, masked as the corresponding 8-aminoquinoline (AQ) amide (**1a**)[41], was selected as the model substrate, and diethyl azodicarboxylate (DEAD) **2a** as the coupling partner (Table 1). After an initial screening of metal catalysts, we were delighted to find that only Cu(OAc)₂ in toluene at 80 °C was able to afford the expected δ-amination product **3a** (the stereochemistry of **3a** was unambiguously confirmed by X-ray crystallography) (Table 1, entry 5). Other representative metal salts catalysts (M = Rh, Ir, Pd, Co, or Ni) did not show any catalytic activity

(Table 1, entries 1–4, 6). After screening of various copper salts catalysts (Table 1, entries 7–13), promising results demonstrated the feasibility of this reaction and proceeded with excellent regio- and stereoselectivity (>20:1 δ:β, >20:1 *E:Z*). Moreover, these results indicated that Cu(I) exhibits significantly higher catalytic efficiency compared to Cu(II). The desired δ-amination product **3a** was obtained in 81% yield by using catalytic amounts of CuCl (10 mol %), and DEAD (2.0 equiv), at 80 °C in toluene for 8 h (Table 1, entry 11). Next, a survey of solvents indicated that 1,2-dichloroethane could be improved to a higher conversion to obtain a 89% yield and that the reaction was sluggish when polar solvents were employed (Table 1, entries 11, 14–17). Additionally, reducing the catalyst loading from 10 to 5 mol % resulted in a slight increase of the yield, albeit requiring a longer reaction time (entries 18 and 19). Then continuing to reduce catalyst loading to 1 mol %, while adjusting the temperature to 90 °C, we obtained a 94% isolated yield for 8 h, and as optimized conditions for this reaction (entry 20). Furthermore, when the reaction was carried out under an Ar atmosphere, almost the same yield as with entry 20 was observed. Finally, we examined the effect of directing groups. Substitution on the arene moiety of AQ (**3b** 5-MeO-AQ (MQ) and **3c** 5-Cl-AQ (CQ)) did not affect the reaction, of note is the inability for alkene substrates, including those with bidentate auxiliary (**3e**), monodentate auxiliary (**3f**) or without any directing group (**3g**), to afford the desired product, with only slight detected using the 2-(pyridin-2-yl)isopropyl (PIP) directing group (**3d**).

### Substrate scope

Under optimal conditions, the substrate range for δ-amination was investigated (Table 2). The reactions showed high regioselectivity and excellent *E/Z* selectivity (>20:1) in all cases. Firstly, we explored the scope of azo compounds (see Supplementary Table 1 for details). It was observed that this reaction is limited to azodicarboxylates, all of which could give corresponding amination products in good yields (**4–7**). For the substrates AIBN and azobenzene, no desired products were observed (**8, 9**). We then explored the scope of alkene substrates using diethyl azodicarboxylate and found that the reaction was compatible with a variety of substituted alkenes. δ-Amination with linear, branched, and cyclic substitutions were obtained in moderate to excellent yields (**10–17**). Alkene substrates bearing a di-α-substituent are suitable for this reaction and mono-α-substituted substrate underwent this transformation in good yield and with >10:1 dr (**14, 15**). Regardless of the aryl ring's electrical properties or replacement patterns, it was well tolerated to generate corresponding products in

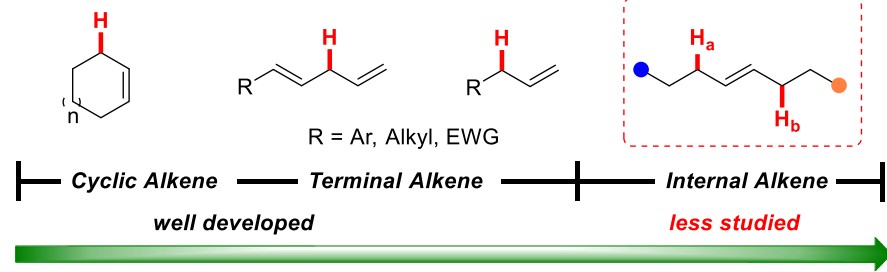

a) Palladium-catalyzed allylic C-H activation

b) Challenges in allylic C−H amination of unactivated internal alkenes

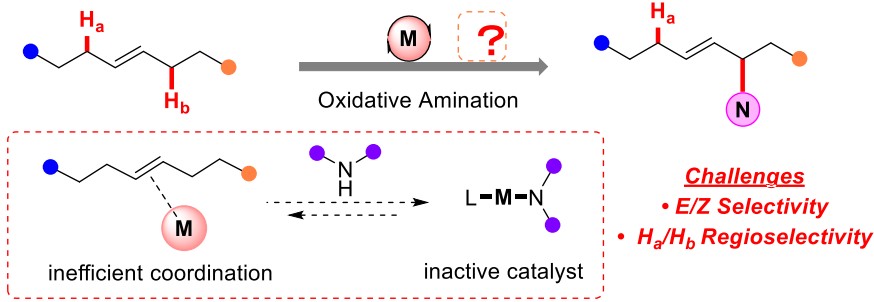

c) *Our strategy*

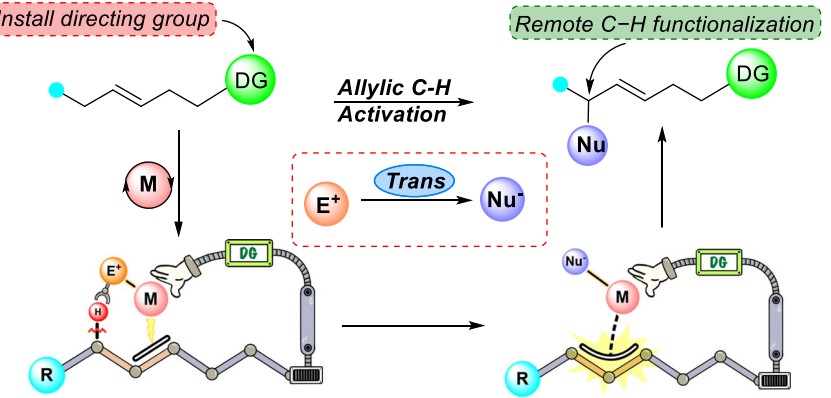

**Fig. 2 | Background of current work and reaction design. a** Palladium-catalyzed allylic C–H activation. **b** Challenges in allylic C−H amination of unactivated internal alkenes. **c** Our strategy.

modest yields. (**18**–**24**). Intriguingly, we also examined the impact of different substituents with long alkyl chains. Functional groups containing oxygen and nitrogen were both well-tolerated (**25**–**29**). In addition, chlorine, iodine, ester, azide and amide groups were also accommodated (**27**–**35**). Subsequently, we turned our attention to N-Boc-protected amino acid substrates. As expected, the desired

amination (**29**) has been smoothly achieved in 65% yield with >10:1 dr. Notably, a substrate bearing a terminal alkyne or terminal alkene moiety was also applicable to this reaction (**33, 34**). Likewise, a range of heterocyclic substrates, including those bearing cyclohexanone (**36**), indole (**37**), N-Boc-piperidine (**38**), and phthalimide (**39**) motifs, were shown to be compatible. Of particular note is that no corresponding

**Table 1 | Optimization of reaction conditions[a,b]**

| entry | cat. | sol. | t (°C) | h | yield (%)[b] |
|---|---|---|---|---|---|
| 1 | [Cp*RhCl$_2$]$_2$ (10 mol%) | Toluene | 80 | 20 | 0 |
| 2 | [Cp*IrCl$_2$]$_2$ (10 mol%) | Toluene | 80 | 20 | 0 |
| 3 | Pd(OAc)$_2$ (10 mol%) | Toluene | 80 | 20 | 0 |
| 4 | Co(acac)$_2$ (10 mol%) | Toluene | 80 | 20 | 0 |
| 5 | Cu(OAc)$_2$ (10 mol%) | Toluene | 80 | 20 | 56 |
| 6 | NiBr$_2$ (10 mol%) | Toluene | 80 | 20 | 0 |
| 7 | Cu(OTf)$_2$ (10 mol%) | Toluene | 80 | 20 | 22 |
| 8 | CuCl$_2$ (10 mol%) | Toluene | 80 | 8 | 58 |
| 9 | CuO (10 mol%) | Toluene | 80 | 8 | 43 |
| 10 | CuI (10 mol%) | Toluene | 80 | 8 | 73 |
| 11 | CuCl (10 mol%) | Toluene | 80 | 8 | 81 |
| 12 | CuOAc (10 mol%) | Toluene | 80 | 8 | 61 |
| 13 | CuTc (10 mol%) | Toluene | 80 | 8 | 72 |
| 14 | CuCl (10 mol%) | THF | 80 | 8 | 75 |
| 15 | CuCl (10 mol%) | MeOH | 80 | 8 | Trace |
| 16 | CuCl (10 mol%) | DCE | 80 | 8 | 89 |
| 17 | CuCl (10 mol%) | MeCN | 80 | 8 | 56 |
| 18 | CuCl (5 mol%) | DCE | 80 | 12 | 91 |
| 19 | CuCl (1 mol%) | DCE | 80 | 15 | 92 |
| 20 | CuCl (1 mol%) | DCE | 90 | 8 | 94 |
| 21[c] | CuCl (1 mol%) | DCE | 90 | 8 | 93 |

3b 73% yield 3c 90% yield 3d 29% yield

3e ND 3f ND 3g ND

[a]Reaction conditions: **1a** (0.40 mmol, 1.0 equiv), **2a** (0.80 mmol, 2.0 equiv), and catalyst (1–10 mol%) in solvent (2.0 mL) under air at the indicated temperature.
[b]Isolated yield.
[c]Under Ar.

products were observed for alkene substrates bearing active reaction sites (red *, Table 2).

Moreover, the range of the Cu/azodiformate cooperative catalytic system can be expanded to include late-stage modifications of medicinal molecules and natural product derivatives. The (+)-citronellal derivatives' C−H amination performed well, affording the corresponding product (**40**) with a yield of 72%. Furthermore, under optimal conditions, other derivatives of natural products, including coumarin (**41**), cholic acid (**43**), dehydrocholesterol (**45**), and erucic acid (**46**), provided the desired products in moderate yields without compromising the functionality of the molecules. The reaction conditions were also applicable to derivatives containing pharmacologically active motifs of chalcone (**42**) and estrone (**44**).

## Synthetic applicability

To further demonstrate the synthetic utility of this C−H amination reaction, a gram-scale reaction was carried out under a lower catalyst loading (reducing catalyst loading from 1 to 0.5 mol %), and the reaction also afforded **3a** in 92% yield albeit extending the reaction time to 36 h (Fig. 3a), which also showed the high efficiency of the Cu/azodiformate catalytic system. The internal alkenes could be reduced to form **47** or oxidized to prepare **48** with K$_2$OsO$_4$. The product **5** was subjected for deprotection (2 M HCl in MeOH,) to afford hydrazinium salt intermediate, which subsequently could be converted to corresponding simple N-Boc allylic amine **49**. In addition, the amination products **5** also could be converted to corresponding δ-amino acid analogue **50** in good overall yield (Fig. 3b).

## Table 2 | Substrate scopes[a,b]

[a]Reaction conditions: **1a** (0.40 mmol, 1.0 equiv), **2a** (0.80 mmol, 2.0 equiv), CuCl (0.004 mmol, 1 mol%), in DCE (2.0 mL) at 90 °C for 8 h.

[b]Isolated yield.

[c]Red * substrates bearing active reaction sites.

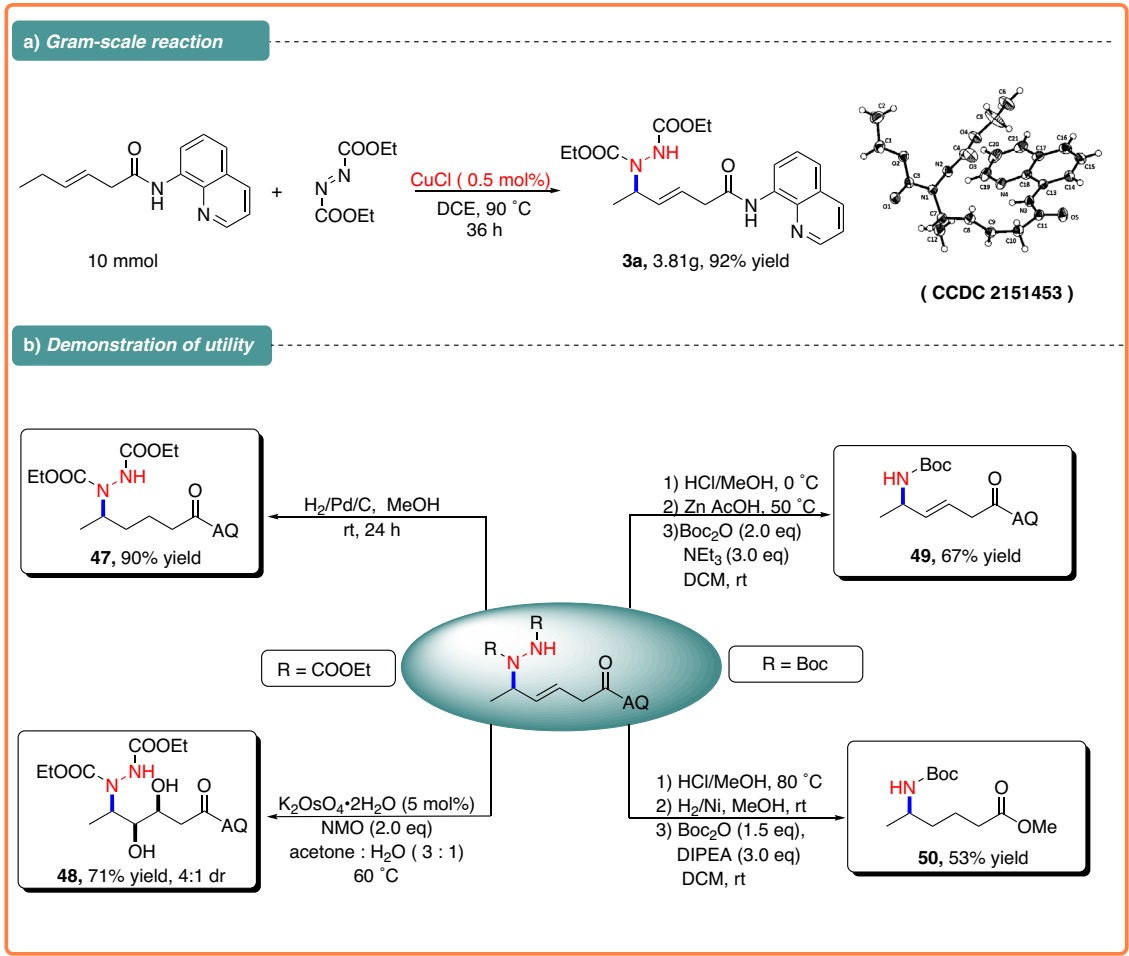

**Fig. 3 | Synthetic applicability. a** Gram-scale reaction. **b** Demonstration of utility.

## Mechanistic study

To explore the underlying mechanism of the allylic C–H amination process, a series of control experiments and isotope labeling experiments were conducted. Intriguingly, reaction of both trans- and cis-hexenamide (**1a** and **51**) gave the same product **3a** in similar yield, indicating that these reactions might proceed via similar π-allyl-Cu intermediates (Fig. 4a). To investigate the potential involvement of radical intermediates in the reaction, radical scavengers such as TEMPO, BHT, and α-cyclopropylstyrene were employed (Fig. 4b). Under the standard conditions, all of these exerted negligible effects on the formation of **3a**, suggesting that the reaction might not proceed in a radical pathway. No reaction was observed with the amide derived from the N-methyl amide **53**, highlighting the key role of the NH chelation from the directing group (Fig. 4c). In addition, the production of the **58**-$d_2$, featuring 98% deuteration at the allylic site, resulting from the reaction of **55**-$d_3$ (>99% D) and **2a**, indicating that the allylic C–H cleavage pathway is almost irreversible. Furthermore, a deuterium labeling experiment using **56**-$d_2$ (59% D) resulted in a 90% yield of **59**-$d_2$ (59% D) without deuterium scrambling. Additionally, it was observed that the deuteration at the internal alkene of **57**-$d_2$ (97% D) and the corresponding product, **60**-$d_2$ (97% D), were identical. This provided convincing proof that the reaction pathway would not involve the alkene isomerization of **1a** (Fig. 4d). Moreover, Fig. 4e and f illustrated that the parallel and intermolecular kinetic isotope effects (KIE) were determined to be 2.6 and 2.7, respectively. All these findings are in accordance with the allylic C–H activation mechanism, suggesting the allylic C–H cleavage might be involved in the turnover-limiting step.

## DFT calculations

To gain more insights into the Cu-catalyzed intermolecular allylic C–H amination of unactivated alkenes with azodiformates, a series of DFT calculations were conducted to further understand the reaction mechanism and origin of regioselectivity (Fig. 5). Indeed, the key role of azodiformate as a base has been proposed for this catalytic system with Cu. First, a deprotonation process of **1a** by **2a** occurs via **TS-1** to afford **B**, followed by dissociation into an active Cu(I) amino intermediate **C** with the Gibbs free energy barrier of 15.6 kcal/mol relative to **A**. From the substrate-coordinated Cu(I) complex **C**, the nitrogen of **2a** re-coordinated to the copper in the Cu(I) intermediate **C**, leading to the formation of complex **D**, which then underwent allylic C–H activation through a Cu/azodiformate-promoted electrophilic C–H cleavage process via **TS-2**, resulting in the formation of the π-allyl-Cu complex **E**. Meanwhile, Cu(I) was oxidized to Cu(III) intermediate **E** via **TS-2**. Subsequently, intermediate **E** with the nucleophilic nitrogen coordinated to Cu(III), facilitating the C–N reductive elimination. The product-coordinated Cu(I) complex **F** eventually liberated the product and regenerated the active catalytic species. With respect to the regioselectivity, two competing transition states of the inner-sphere pathway were explored by DFT calculations. The δ-amination pathway through **TS-3** was found to be at least 4.9 kcal/mol more favorable than the β-amination pathway. According to the transition-state model (Fig. 5b), the β-amination pathway through **TS-3'** encounters a notable challenge in attacking the more internal site. This difficulty arises from the need to surmount increased steric hindrance between **2a** and the side chain of **1a**. Consequently, this steric hindrance prompts the displacement of Cu away from the active region, rendering **TS-3'**

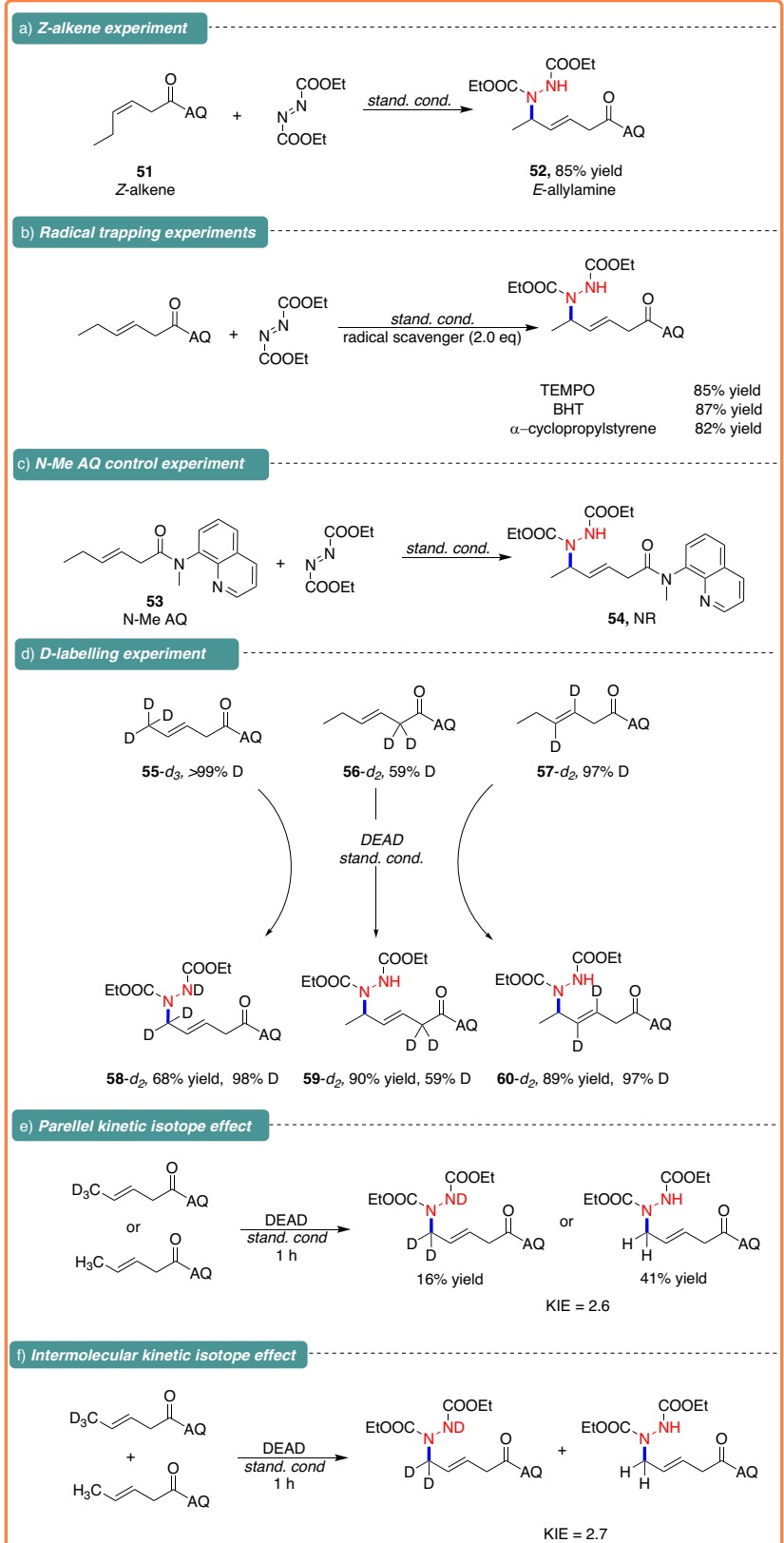

**Fig. 4 | Mechanistic experiment. a** *Z*-alkene experiment. **b** Radical trapping experiments. **c** N-Me AQ control experiment. **d** D-labeling experiment. **e** Parellel kinetic isotope effect. **f** Intermolecular kinetic isotope effect.

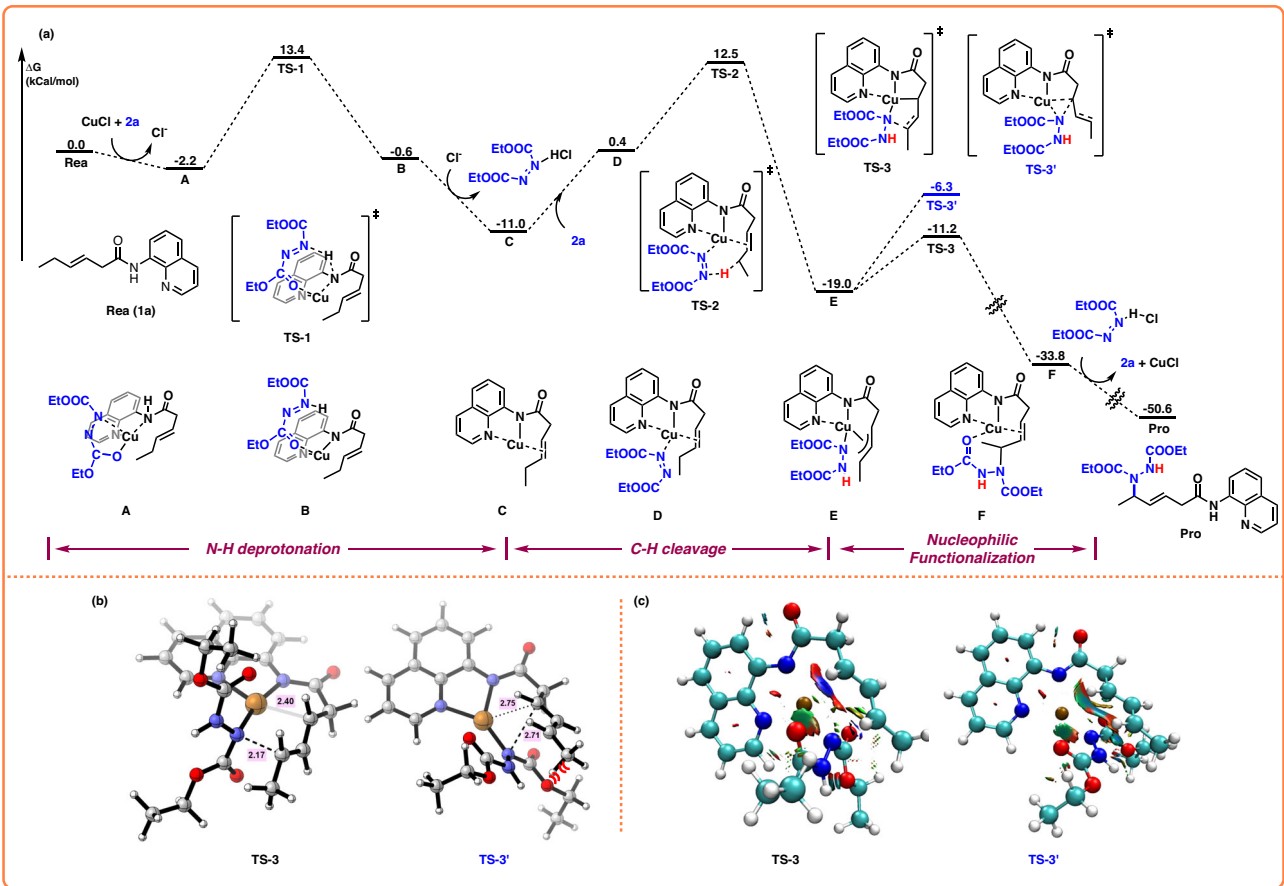

**Fig. 5 | DFT calculations. a** Free energy changes of the favorable pathway. **b** The transition states for **TS-3** or **TS-3′** elimination process. **c** NCI analysis of the weak non-covalent interactions in **TS-3** or **TS-3′**.

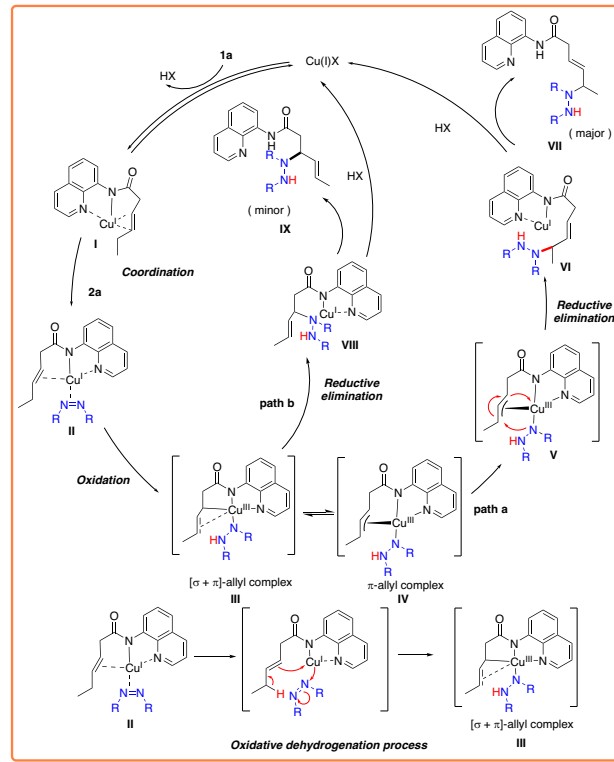

**Fig. 6 |** Proposed reaction mechanism.

energetically disfavored. As depicted in Fig. 5c, NCI analysis highlights the pivotal role of the Cu–C interaction in the formation of the C–N bond, in which the Cu–C interaction strength in **TS-3** surpasses that in **TS-3′**. Additionally, **TS-3′** exhibits heightened and extensive steric hindrance between **2a** and the side chain of **1a**. These two factors collectively contribute to the energetic preference for the δ-amination pathway through **TS-3**. The rate-determining step involved C–H bond activation via **TS-2**, with an overall barrier of 23.5 kcal/mol, in agreement with the experimental kinetic studies (Fig. 4e, f).

## Proposed reaction mechanism

In consideration of the aforementioned experimental findings and relevant literature, we have put forth a plausible mechanism for the Cu-catalyzed selective allylic C–H amination, illustrated in Fig. 6. Initially, copper(I) formed a complex **I** with the nitrogen atom of the directing group, positioning it in proximity to olefin **1a**. The close proximity enhanced the π-Lewis acid activation of the olefin, followed by an oxidation process to generate Cu(III) complex **III**. The key to this reactivity is diethyl azodicarboxylate that shows the effect of oxidative dehydrogenation and promotes Cu-mediated C–H cleavage to generate [σ + π]-allyl copper(III) species **IV**. The newly formed [σ + π]-allyl complex **IV** could react under two competitive pathways. Based on the computational studies, the [σ + π]-allyl complex **IV** prefers to undergo C–N reductive elimination via path **a** to obtain the δ-amination product and provide regenerated CuCl for the next cycle.

## Discussion

In summary, we have reported a Cu-catalyzed allylic C–H activation method for the direct intermolecular amination of unactivated internal

olefins with azodiformates. The removable 8-aminoquinoline directing group dictated the regioselectivity and stabilized the π-allyl-copper intermediate. We found that azodiformates served as unique oxidants, interacting with copper to facilitate C−H cleavage through an oxidative dehydrogenation process. This operationally simple method proceeded with variety of alkyl- and aryl-substituted olefins bearing various functional groups and exhibited high regioselectivity and stereoselectivity. Future research will aim to gain insight into the reaction mechanism and broaden the range of nucleophiles. Reports on these findings will be provided at a later date.

## Methods
### General procedure
A mixture of amide (0.40 mmol, 1 equiv), CuCl (0.004 mmol, 0.01 equiv), and azodicarboxylate (0.80 mmol, 2.0 equiv) in DCE (2.0 mL) in a 10 mL glass vial (sealed with PTFE cap) was heated at 90 °C for indicated time. The reaction progress was monitored by thin layer chromatography. Upon completion, the reaction mixture was concentrated in vacuo and purified by silica gel column chromatography to afford the desired products.

## Data availability
The data generated in this study are provided in the Supplementary Information file. The experimental procedures, data of NMR, and HRMS have been deposited in Supplementary Information file. The X-ray crystallographic coordinates for structures reported in this study have been deposited at the Cambridge Crystallographic Data Centre (CCDC: 2151453). These data could be obtained free of charge from The Cambridge Crystallographic Data Centre (https://www.ccdc.cam.ac.uk/data_request/cif). Source data are present. All data are available from the corresponding author upon request. Source data are provided with this paper.

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

## Acknowledgements

This work was supported by the National Key Research & Development Program of China (2021YFF0701800 and 2022YFC2703400), the Fundamental Research Funds for the Central Universities (YG2022ZD021), NSFC (No. 22071147), and STCSM (22ZR1435100).

## Author contributions

The project was conceived and directed by S.-Y.Z., L.W. designed the experiments and analyzed the data. Z.-H.L. performed the DFT calculations. L.W., C.-L.W., P.-F.L., J.-C.K. and J.Z. performed the experiments. L.W., Y.H., R.-X.L. and H.-Y.B. prepared the manuscript. All authors discussed the results and commented on the manuscript.

## Competing interests

The authors declare no competing interests.
