## [Peer Review File · Nature Communications]

Cooperative Cu/Azodiformate System-Catalyzed Allylic C–H
Amination of Unactivated Internal Alkenes Directed by
AminoquinolineREVIEWER COMMENTS

Reviewer #1 (Remarks to the Author):

The submission by Zhang et al. describes a challenging Cu-catalyzed allylic C–H amidation of internal alkenes. The success of this reaction critically relies on strategic substrate design. The reaction is guided by a removable bidentate directing group and proceeds through the π -allyl-metal intermediate to effect C–H functionalization at distal δ -position selectively. For the source of nitrogen functionality, the readily available azodicarboxylates were employed, and they exhibited dual roles of oxidant and reagent. The protocol is operationally simple, accommodates a range of substrates including bioactive scaffolds, and products were isolated in high yields with E-selectivity. Adequate mechanistic studies were conducted along with the DFT study to support the reaction mechanism. Overall, the key finding is unique, and I recommend the manuscript for publication in Nature Communications after the following modifications.

The >10:1 dr for substrates 14 and 15 should be indicated in Table 2.

Is this reaction limited to only diazocarboxylates? The outcome in the case of azobisisobutyronitrile (AIBN) can be included.

In Table 1 (entries 1, 3, 4, and 6): these reactions did not furnish the desired product. Is the starting material recovered here? Any side product, if detected, can be specified for a comprehensive understanding.

Compounds 25, 28, 33, 37 are not very pure. Clean NMR spectra may be provided.

Compound 35: ^{19}F NMR is missing.

Compound 44: the carbonyl signal of estrone is missing in spectra.

The melting point of solid new compounds can be reported.

Reviewer #2 (Remarks to the Author):

The author presented a regioselective Cu-catalyzed oxidative allylic C(sp³)-H amination of internal olefins with azodiformates in the manuscript, achieving good regioselectivity through the use of a bidentate directing group. However, the mechanistic calculations are too crude to elucidate the proton transfer at the amide position in their computational model, provide an unclear explanation for the role of the substrate, and lack discussions on the key factors influencing regioselectivity. Without these details would hinder chemists for further comprehending and expanding upon the reaction. Therefore, I am not unable to consider publish the manuscript.

Major:

1. The role of substrate 2a in the reaction mechanism is indeed crucial. Based on the author's computational model, when substrate 1a coordinates with Cu, the N-H bond at the amide position is cleaved. Since no base is introduced into the reaction system, is it plausible to consider that substrate 2a may act as a base, facilitating deprotonation in this step? However, in the mechanism illustrated in Figure 5, substrate 2a appears to exhibit redox properties when it abstracts a hydrogen atom from substrate 1a, causing an increasing oxidation state of Cu. This implies a different functional role for substrate 2a in this particular step. It's essential to provide a more detailed explanation that can clarify why substrate 2a performs different functions in these distinct stages of the reaction.

2. In Figure 5, authors propose a transition from Cu(I) to Cu(III) in the mechanism. This is somewhat assertive, and whether there is experimental evidence to confirm the existence of Cu(III) species in the system.

3. The mechanism depicted in Figure 5 is the crucial step determining regioselectivity. The author should provide a detailed analysis based on the transition state structures, examining the nature of hydrogen which is transferred and highlighting the key interactions. Such information would be beneficial for chemists understanding this reaction.

4. In Figure 4, the author determined the kinetic isotope effect (KIE) values in experiment for this reaction. To further validate the correctness of the mechanism, the author can also calculate the theoretical KIE values and compare it with the experimental results.

Minor:

1. In Table 2, the author uses a red * to mark the active sites. It is possible that the active site for structure 37 is labeled incorrectly.

2. In Figure 3b, the description mentions using ethyl acetate (EA) as the solvent for the reaction in the upper right corner. However, there is a discrepancy in the text, where it states that methanol (MeOH) is the solvent at line 162

3. In Figure 4d, in the deuterium-labeled experiment, based on the mechanism described in the text, the N-H in the structure 58-d2 should be N-D.

Reviewer #3 (Remarks to the Author):

The manuscript entitled "Cooperative Cu/Azodiformate System-Catalyzed Allylic C-H Amination of Unactivated Internal Alkenes" reports a new reaction system for the regio- and stereoselective allylic C-H amination with the azo dicarboxylates. The reaction mechanism is very unique, and the azo dicarboxylate worked as initially electrophile and finally nucleophile. In particular, the C-H cleavage process including Cu, allylic substrate, and azo dicarboxylate deserves significant attention.

However, this reviewer has several critical concerns. First, the authors state that this is the allylic C-H amination of "unactivated" internal alkenes, but the substrate is truly "unactivated"? The substrate with the aminoquinoline (AQ) directing group is an apparently "activated" alkene. Of course, this is the first successful application of AQ in the regioselective allylic C-H amination, but the novelty cannot meet the standard level of Nat. Commun. because of here are many precedents of the regio- and stereochemical functionalizations of AQ-containing alkene substrates. At least, the catalytic asymmetric version should be developed.

On the basis of the above considerations, I do think that this submission is the borderline case. If the authors are willing to address the aforementioned critical problems, I can support acceptance of this paper.

Additional comments:

- 1) In the TOC graphic, the word "dehydrogenation" should be replaced with "C-H cleavage".
- 2) Page 2, final line; "direct oxidation of allyl..." should be replaced with "oxidative allylic...".
- 3) In Figure 2c, "Region-selective" should be replaced with "Regio-selective". Additionally, the molecular orientation of starting substrate and product should be consistent with the illustrations.
- 4) In Table 1, is the reaction generally performed under N₂ or air? The atmospheric conditions should also be shown in the caption.

- 5) Page 5, final part seems to be unfair, at least, for me. The authors mentioned the traditional Cu-catalyzed allylic substitutions of allyl (pseudo)halides with organometallic reagents, which are totally different and have nothing to do with the present reaction.
- 6) In Table 2, compound 15; the dr value should be shown.
- 7) In Figure 3, compound 48; the dr value should be shown.
- 8) Page 9; there are the grammatical errors in the first sentence.

December 12, 2023

List of responses

1. Responses to reviewer 1:

1) Comment #1: “The dr for substrates 14 and 15 should be indicated in Table 2.”

Our Response: Thanks for your suggestions. For substrate **14**, due to the presence of only one chiral center in the product, there is no diastereoselectivity. As for substrate **15**, the Dr value has been annotated in Table 2.

2) Comment #2: “Is this reaction limited to only diazocarboxylates? The outcome in the case of azobisisobutyronitrile (AIBN) can be included.”

Our Response: Thanks for your suggestions. Upon screening a wide range of diazo compounds, it was observed that this reaction is limited to diazocarboxylates, with no reactivity observed for other diazo compounds, including AIBN (azobisisobutyronitrile). Given the limitations of the substrate table, we will not include the example of AIBN in Table 2.

3) Comment #3: “In Table 1(entries 1, 3, 4, and 6): these reactions did not furnish the desired product. Is the starting material recovered here? Any side product, if detected, can be specified for a comprehensive understanding.”

Our Response: Thanks for your suggestions. In Table 1 (entries 1, 2, 3, 4, and 6), the employed metal catalysts exhibited almost no reactivity, resulting in a significant excess of starting material **1a**, without any observed generation of other by-products. Furthermore, the starting material is recoverable under these conditions.

4) Comment #4: “Compounds 25, 28, 33, 37 are not very pure. Clean NMR spectra may be provided; Compounds 35: ^{19}F NMR is missing; Compounds 44: the carbonyl signal of estrone is missing in spectra; The melting point of solid new compounds can be reported.”

Our Response: Thanks for your suggestions. We have carried out a secondary purification of compounds **25**, **28**, **33**, and **37**; The ^{19}F NMR of compound **35** has been completed, as well as the carbonyl signal of estrone in compound **44**; The melting points of solid products in the reaction were measured, and all these modifications have been annotated in the supporting information.

2. Responses to reviewer 2:

1) Comment #1: “The role of substrate 2a in the reaction mechanism is indeed crucial. Based on the author’s computational model, when substrate 1a coordinates with Cu, the N-H bond at the amide position is cleaved. Since no base is introduced into the reaction system, is

it plausible to consider that substrate **2a** may act as a base, facilitating deprotonation in this step? However, in the mechanism illustrated in Figure 5, substrate **2a** appears to exhibit redox properties when it abstracts a hydrogen atom from substrate **1a**, causing an increasing oxidation state of Cu. This implies a different functional role for substrate **2a** in this particular step. It's essential to provide a more detailed explanation that can clarify why substrate **2a** performs different functions in these distinct stages of the reaction."

Our Response: Thanks for your suggestions. Substrate **2a** plays a dual role in the reaction mechanism, functioning both as an oxidizing agent and a nitrogen source. According to the reaction mechanism, upon the coordination of substrate **1a** with copper, the N-H bond at the amide position undergoes cleavage. Due to the absence of an introduced base in the reaction system, we postulate that substrate **2a** may function as a base. We contend that its involvement in promoting deprotonation during this step is reasonable, as there is literature evidence suggesting that azodiformates can act as bases for hydrogen abstraction (*J. Am. Chem. Soc.* **1966**, *88*, 2328; *J. Org. Chem.* **1967**, *32*, 727; *J. Am. Chem. Soc.* **2008**, *130*, 14048). Based on the hypothesis that substrate **2a** serves as a base, along with an excess of **2a**, we reintroduced the first deprotonation step in the DFT calculation outlined in Figure 5. Facilitated by copper, substrate **2a** efficiently abstracts proton from substrate **1a** with an energy barrier of 15.6 kcal/mol. Following the extraction of a hydrogen atom from substrate **1a**, substrate **2a** assumes the role of an oxidant, resulting in an increased oxidation state of copper. It has been reported in certain literature that diazo compounds can function as oxidants (*Tetrahedron Lett.* **2009**, *50*, 1493; *J. Am. Chem. Soc.* **2018**, *140*, 1612).

2) Comment #2: "In Figure 5, authors propose a transition from Cu(I) to Cu(III) in the mechanism. This is somewhat assertive, and whether there is experimental evidence to confirm the existence of Cu(III) species in the system."

Our Response: Thanks for your suggestions. Reductive elimination of Cu(III) intermediates is often proposed as a key step in many copper-catalyzed or -mediated C-C or C-heteroatom bond-forming reactions. However, there still lacks concrete evidence on this key step, mainly because Cu(III) complexes are usually too unstable to be isolated and structurally characterized. To validate the proposed transition from Cu(I) to Cu(III) in the mechanism, an electron paramagnetic resonance (EPR) experiment was performed to verify whether there was a Cu(II) species during the reaction process. However, the results showed no visible EPR signals corresponding to Cu(II) species. However, we consider the reaction to be an oxidation-reduction type, making it less likely to be catalyzed by a Lewis acid. Therefore, we ultimately propose a transition from Cu(I) to Cu(III) in the mechanism.

3) Comment #3: "The mechanism depicted in Figure 5 is the crucial step determining regioselectivity. The author should provide a detailed analysis based on the transition state structures, examining the nature of hydrogen which is transferred and highlighting the key interactions. Such information would be beneficial for chemists understanding this reaction."

Our Response: We thank the reviewer for this comment. We have provided an analysis based on transition state structures in the revised manuscript to clarify the regioselectivity. We contend that, during the hydrogen transfer process, the hydrogen atom itself undergoes no inherent property changes. The coordination of copper with diazocarboxylates enhances the

basicity of nitrogen atoms, thereby facilitating deprotonation.

4) Comment #4: “In Figure 4, the author determined the kinetic isotope effect (KIE) values in experiment for this reaction. To further validate the correctness of the mechanism, the author can also calculate the theoretical KIE values and compare it with the experimental results.”

Our Response: Thanks for your suggestions. At the same theory level, we have calculated the reaction in KIE experiment and used freqchk script in Gaussian to analyze the result of deuterated reaction. The theoretical KIE values is 3.34, which is close to the experimental results.

5) Comment #5: “In Table 2, the author uses a red * to mark the active sites. It is possible that the active site for structure 37 is labeled incorrectly.”

Our Response: Thanks for your suggestions. Through a literature survey, it was found that both the 2nd and 3rd positions of indole are susceptible to undergo substitution reactions with diazocarboxylates (eg: Tetrahedron **2005**, *61*, 2401–2405; ACS Catal. **2022**, *12*, 7511–7516).

6) Comment #6: “In Figure 3b, the description mentions using ethyl acetate (EA) as the solvent for the reaction in the upper right corner. However, there is a discrepancy in the text, where it states that methanol (MeOH) is the solvent at line 162; In Figure 4d, in the deuterium-labeled experiment, based on the mechanism described in the text, the N-H in the structure 58-d2 should be N-D.”

Our Response: Thanks for your suggestions. In the revised manuscript, the notation in Figure 3b indicating the use of ethyl acetate (EA) as the reaction solvent in the upper right corner has been corrected to methanol (MeOH). Additionally, the N-H in compound **58-d₂** has been revised to N-D in the deuterium-labeled experiment.

3. Responses to reviewer 3:

1) Comment #1: “First, the authors state that this is the allylic C-H amination of “unactivated” internal alkenes, but the substrate is truly “unactivated”? The substrate with the aminoquinoline (AQ) directing group is an apparently “activated” alkene.”

Our Response: Thanks for your suggestions. The reactivity of intramolecular olefin is much lower than that of terminal olefin. Generally, in transition metal catalyzed reactions, the activity order of alkenes: straight chain alpha-olefin (terminal olefin) > cyclohexene > branched-chain alpha-olefin > Z- internal alkene > E-internal alkene. In this regard, directing groups (AQ) are assembled on olefin substrates with the aim of enhancing the coordination effects between olefin and copper catalysts to assist in achieving the activation of the olefin. However, it cannot be asserted that the sole installation of the AQ directing group in the substrate will lead to the transformation into an activated alkene. The term "AQ-directed unactivated alkene" has been documented in some literature, for example: (J. Am. Chem. Soc. **2019**, *141*, 18475–18485; J. Am. Chem. Soc. **2019**, *141*, 10048–10059; J. Am. Chem. Soc. **2018**, *140*, 3542–3546; J. Am. Chem. Soc. **2017**, *139*, 11261–11270; J. Am. Chem. Soc. **2016**, *138*, 14705–14712; J. Am. Chem. Soc. **2021**, *143*, 1195–1202; J. Am. Chem. Soc. **2018**, *140*, 16929–16935).

2) Comment #2: “At least, the catalytic asymmetric version should be developed.”

Our Response: We are well aware that asymmetric catalysis is currently a research hotspot. Our initial concept was to develop a catalytic asymmetric method for constructing chiral C-N bonds, with the aim of elevating the quality of our manuscripts. Following a year of systematic experimental screening, the highest enantioselectivity we achieved was 25%. Consequently, we made the decision to temporarily halt chiral screening, and we sincerely regret this choice. Due to the unsatisfactory results, we have opted against including this segment of the screening work in the manuscript. However, in the following sections, I will present a subset of the screening results for the reviewer's consideration, with the expectation that valuable insights can be provided.

Table 1

In table 1, CuCl was employed as the catalyst for screening various chiral ligands (some other ligands are not listed in the table 1). It was observed that only the (R)-BINAP ligand exhibited a 6% ee value. However, the remaining ligands failed to exert chiral control.

Table 2

entry	cat	ligand	t	ee
1	CuCN	Ligand A	12h	2%
2	Cu(OAc) ₂	Ligand A	12h	2%
3	Cu(OH) ₂	Ligand A	12h	2%
4	Cu(OTf) ₂	Ligand A	12h	0%
5	Cu(CH ₃ CN) ₄ PF ₆	Ligand A	12h	5%
6	CuTc	Ligand A	12h	11%
7	Cu(acac) ₂	Ligand A	12h	0%
8	CuI	Ligand A	12h	4%
9	CuCl	Ligand A	12h	6%

In Table 2, (R)-BINAP was selected as the chiral ligand, and a screening of copper metal salts catalysts was conducted. The results indicate that CuTc, in comparison to other copper catalysts, achieved an 11% ee value.

Table 3

 ee : 11%	 Ar = 4-Me-Ph ee : 11%	 Ar = 3,5-Me-Ph ee : 7%	 ee : 7%	 ee : 3%
 ee : 8%	 ee : 15%	 Ar = 3,5-t-Bu-4-OMe-Ph ee : 2%	 ee : 0%	 ee : 0%
 ee : 0%	 ee : 0%	 ee : 19%	 ee : 25%	 ee : 2%
 ee : 2%	 ee : 20%	 ee : 11%	 ee : 13%	 ee : 15%
 ee : 7%	 ee : 0%			

In Table 3, CuTc was employed as the catalyst for a systematic screening of phosphine ligands. It is evident that the majority of phosphine ligands exhibited enantioselectivity; however, the results were somewhat unsatisfactory. Only the (R,R)-(-)-2,3-bis(*t*-butylmethylphosphino)quinoxaline ligand demonstrated a 25% ee value.

Table 4

entry	cat (10mol%)	Sol	t	yield	ee
1	CuTc	Toluene	12h	67%	25%
2	CuTc	DCM	12h	71%	4%
3	CuTc	THF	12h	57%	16%
4	CuTc	Et ₂ O	12h	45%	7%
5	CuTc	MeCN	12h	41%	4%
6	CuTc	DCE	12h	68%	14%
7	CuTc	Acetone	12h	52%	17%
8	CuTc	DMF	12h	39%	13%
9	CuTc	DMSO	12h	48%	5%
10	CuTc	1,4-dioxane	12h	54%	20%
11	CuTc	EA	12h	38%	5%
12	CuTc	CH ₃ OH	12h	13%	2%
13	CuTc	Chlorobenzene	12h	65%	13%
14	CuTc	Benzotrifluoride	12h	58%	18%
15	CuTc	Mesitylene	12h	66%	20%

In Table 4, CuTc was selected as the catalyst, and (R,R)-(-)-2,3-Bis(*t*-butylmethylphosphino)quinoxaline was chosen as the chiral ligand. Solvent screening was conducted, revealing that toluene exhibited comparatively favorable results.

Subsequently, we conducted screenings on substrate ratios, reaction temperatures, and additives, but no breakthroughs were attained. The optimal outcome achieved was a 25% enantiomeric excess. The following exploratory experiments were conducted, and an analysis of the screening results was performed.

Table 5

In Table 5, preformed complexes **A** and **B** were prepared. Under standard conditions, substrate **1a** was introduced into the system. After several hours, the appearance of ligand **E** and ligand **F** on the TLC plate was distinctly observed. This result indicates an excessively strong coordinating ability of AQ, leading to ligand exchange and subsequent dissociation of the chiral ligand, thereby preventing the establishment of a chiral environment. We conducted an analysis of the system. Firstly, we identified the limited coordination ability of copper ions. Additionally, there are numerous substrate moieties available for coordination within the system, including the two nitrogen atoms on AQ in substrate **1a** and the two nitrogen atoms in azodiformates. The introduction of a chiral ligand further exacerbates the issue, as there are too many compounds within the system that can coordinate with copper. This leads to a competitive coordination scenario, making it challenging to maintain a chiral reaction environment.

Despite the unsatisfactory outcomes of the catalytic asymmetric version, the introduction of chiral auxiliaries for asymmetric induction is a well-established method in the field of organic synthesis and has been encouraged by the work of Meyer and Yu. Envisaging the potential to achieve diastereoselectivity in allylic C-H amination through the use of a chiral oxazoline amide as a directing group, we conducted a series of screenings. The results are as follows:

Table 6

In Table 6, we initially investigated substituents with varying steric hindrance on the oxazoline ring, including methyl, phenyl, isopropyl, tert-butyl, and indanyl groups. Gratifyingly, the phenyl group demonstrated a high diastereoselectivity (>15:1) and a moderate yield. Conversely, the tert-butyl and indanyl groups were found to suppress the reaction due to their substantial steric hindrance effects. Subsequently, we expanded the substrate scope by introducing alkyl chain R groups. In the table, only a few representative substrates are included. The results indicate that none of them exhibited a diastereoselectivity greater than 5:1. This

observation suggests that steric hindrance from the R group influences the control of enantioselectivity, leading to incompatibility between the chiral directing group and the R group.

Prior to submission, we made the decision to exclude the section on chiral screening from the manuscript as we did not achieve satisfactory enantioselectivity. Subsequently, an in-depth analysis and exploration were conducted on this aspect of the work. We attribute the difficulty in achieving high ee value to the presence of an AQ-directing group in the substrate. We have also attempted to remove the AQ-directing group and transform it into a chiral ligand for screening in allylic C-H amination reactions. The screening process is still ongoing. Finally, we also hope that the reviewers can provide guidance and suggestions on the aforementioned screening results.

3) Comment #3: “In the TOC graphic, the word “dehydrogenation” should be replaced with “C-H cleavage; Page 2, final line; “direct oxidation of allyl...” should be replaced with “oxidative allylic...; In Figure 2c, “Region-selective” should be replaced with “Regio-selective”; Additionally, the molecular orientation of starting substrate and product should be consistent with the illustrations.”

Our Response: Thanks for your suggestions.

We changed “dehydrogenation”(in the TOC graphic) to “C-H cleavage”, changed “direct oxidation of allyl...”(in Page 2, final line) to “oxidative allylic...”, changed “Region-selective”(in Figure 2c) to “Regio-selective”, changed the molecular orientation of starting substrate and consistent with the illustrations.

4) Comment #4: “In Table 1, is the reaction generally performed under N₂ or air? The atmospheric conditions should also be shown in the caption.”

Our Response: Thanks for your suggestions. The reaction proceeds successfully under both N₂ and air conditions. In Table 1, entry 21, screening under Ar conditions has been conducted. Atmospheric conditions have been indicated in the caption.

5) Comment #5: “Page 5, final part seems to be unfair, at least, for me. The authors mentioned the traditional Cu-catalyzed allylic substitutions of allyl (pseudo)halides with organometallic reagents, which are totally different and have nothing to do with the present reaction.”

Our Response: Thanks for your suggestions. We have discussed this matter and agree with the reviewer's opinion, considering that the relevance is not significant. We have already removed the relevant content in the revised manuscript.

6) Comment #6: “In Table 2, compound 15; the dr value should be shown; In Figure 3, compound 48; the dr value should be shown; Page 9; there are the grammatical errors in the first sentence.”

Our Response: Thanks for your suggestions. For compounds **15** and **48**, their Dr values have been annotated in the revised manuscript. The grammatical error in the first sentence on page 9 has been corrected in the revised manuscript.

REVIEWER COMMENTS

Reviewer #1 (Remarks to the Author):

The authors have addressed the majority of the concerns raised by the reviewer. However, it is crucial to transparently disclose unsuccessful results, especially those closely tied to reaction design or development. The mention of an unsuccessful reaction with AIBN raises concerns about the potential limitation of this reaction to only diethyl azodicarboxylate. It is advisable to investigate other azodicarboxylate derivatives and provide clarity on the scope of the reaction. I recommend proceeding with the publication of this work after incorporating these revisions.

Reviewer #2 (Remarks to the Author):

The authors have addressed all my concerns. Then I will support it to publish.

Reviewer #3 (Remarks to the Author):

I have reviewed the original submission of this manuscript. The authors carefully revised it according to comments by reviewers, but my concerns, in particular the development of asymmetric catalysis, still cannot be fully addressed. Thus, I leave the adjudgment on the acceptance to editor.

Just a comment:

The word "aminoquinoline-directed" should be involved in the manuscript title.

January 8, 2024

List of responses

The main corrections in the paper and the responds to the reviewers' comments are as follows:

1. Responses to reviewer 1:

1) Comment #1: "It is crucial to transparently disclose unsuccessful results, especially those closely tied to reaction design or development. The mention of an unsuccessful reaction with AIBN raises concerns about the potential limitation of this reaction to only diethyl azodicarboxylate. It is advisable to investigate other azodicarboxylate derivatives and provide clarity on the scope of the reaction."

Our Response: Thanks for your suggestions. We have conducted screening on the substrate scope of azo compounds, and the results of the screening are presented in the Supporting Information.

Upon screening a wide range of azo compounds, it was observed that this reaction is limited to azodicarboxylates. We postulate that azodicarboxylates may exhibit a better compatibility with the oxidative-reductive properties of copper within the reaction system. With respect to the substrate AIBN, decomposition may occur under high-temperature condition.

2. Responses to reviewer 3:

1) Comment #1: "The word "aminoquinoline-directed" should be involved in the manuscript title."

Our Response: Thanks for your suggestions. We have included the term "aminoquinoline-directed" in the title of the revised manuscript.